# Interaction of Geopolymer Filler and Alkali Molarity Concentration towards the Fire Properties of Glass-Reinforced Epoxy Composites Fabricated Using Filament Winding Technique

**DOI:** 10.3390/ma15186495

**Published:** 2022-09-19

**Authors:** Mohammad Firdaus Abu Hashim, Meor Ahmad Faris, Md Azree Othuman Mydin, Che Mohd Ruzaidi Ghazali, Yusrina Mat Daud, Mohd Mustafa Al Bakri Abdullah, Farah Farhana Zainal, Muhammad Faheem Mohd Tahir, Heah Cheng Yong, Morteza Khorami

**Affiliations:** 1Center of Excellence Geopolymer and Green Technology (CEGeoGTech), Universiti Malaysia Perlis, Perlis 01000, Malaysia; 2Faculty of Mechanical Engineering Technology, Universiti Malaysia Perlis, Kampus Tetap Pauh Putra, Perlis 02600, Malaysia; 3School of Housing, Building and Planning, Universiti Sains Malaysia, Penang 11800, Malaysia; 4Faculty of Ocean Engineering Technology and Informatic, Universiti Malaysia Terengganu, Kuala Nerus, Terengganu 21030, Malaysia; 5Faculty of Chemical Engineering Technology, Universiti Malaysia Perlis, Perlis 01000, Malaysia; 6Civil Engineering Department, Faculty of Engineering, Sriwijaya University, Indralaya 30662, Indonesia; 7Department of the Built Environment, Faculty of Engineering & Computing, Sir John Laing Building, Coventry University, Coventry CV1 5FB, UK

**Keywords:** geopolymer, filament winding, glass-reinforced epoxy, mechanical properties, flame test

## Abstract

This paper aims to find out the effect of different weight percentages of geopolymer filler in glass-reinforced epoxy pipe, and which can achieve the best mechanical properties and adhesion between high calcium pozzolanic-based geopolymer matrices. Different weight percentages and molarities of epoxy hardener resin and high calcium pozzolanic-based geopolymer were injected into the glass fiber. By manually winding filaments, composite samples were produced, and they were then allowed to cure at room temperature. To determine how well the geopolymer matrices adhere to the fiber reinforcement, the microstructure of the composites’ surfaces and perpendicular sections were examined. Maximum values of compressive strength and compressive modulus were 94.64 MPa and 2373.58 MPa, respectively, for the sample with a weight percentage of filler loading of 30 wt% for an alkali concentration of 12 M. This is a relatively wide range of geopolymer weight percentage of filler loading from 10 wt% to 40 wt%, at which we can obtain high compressive properties. By referring to microstructural analysis, adhesion, and interaction of the geopolymer matrix to glass fiber, it shows that the filler is well-dispersed and embedded at the fiber glass, and it was difficult to determine the differences within the range of optimal geopolymer filler content. By determining the optimum weight percent of 30 wt% of geopolymer filler and microstructural analysis, the maximum parameter has been achieved via analysis of high calcium pozzolanic-based geopolymer filler. Fire or elevated temperature represents one of the extreme ambient conditions that any structure may be exposed to during its service life. The heat resistance or thermal analysis between glass-reinforced epoxy (GRE) pipe and glass-reinforced epoxy pipe filled with high calcium pozzolanic-based geopolymer filler was studied by investigating burning tests on the samples, which shows that the addition of high calcium pozzolanic-based geopolymer filler results in a significant reduction of the melted epoxy.

## 1. Introduction

Improvement studies on geopolymer resulted in a variety of new materials with strong applications which are not only meant for cement replacement. In recent years, there has been scientific interest in geopolymer materials due to two major advantages: lower carbon dioxide (CO_2_) emissions than ordinary Portland cement (OPC) and low energy consumption [1]. Geopolymer materials are a product of inorganic polymerization. The term geopolymer was first introduced by Prof. Joseph Davidovits in 1978. Davidovits claimed that an alkaline liquid could be used to react with the silicon (Si) and the aluminum (Al) in a source material of geological origin or in by-product materials such as fly ash, metakaolin, kaolin, and rice husk ash to produce binders [2]. However, a 1908 patent of Kuhl was recognized as the first on the alkali activation of aluminosilicate precursors in order to obtain an ordinary Portland cement (OPC) alternative material [3]. Previous researchers mentioned that the sialate nomenclature of Davidovits “implies certain aspects of the geopolymer gel structure which do not correspond to reality” [4,5]. Through a period, several researchers aimed to utilize geopolymers for the numerous possible uses for these materials, and efforts were made toward the formation of inorganic geopolymers. The literature has several publications on the synthesis, characteristics, and uses of geopolymers [6,7]. Innovative technology, known as geopolymerization, may turn various aluminosilicate minerals into useful items, known as geopolymers or inorganic polymers. Mineral polymers, also known as polysiatate or geopolymers, are among the remaining inorganic chemistry advances made through geopolymerization. These materials offer a significant opportunity to produce composite materials with ideal fire resistance, non-toxic fumes and smokes, and resistance to all organic solvents, in addition to having excellent mechanical properties such as being lightweight and high in strength [8,9,10]. Since clay materials contribute to outstanding mechanical qualities of either thermoplastic or thermoset matrix, composites based on clay have received more attention in research [11,12,13].

A non-metallic pipe system, such as glass fiber-reinforced plastics (GRP) pipes and non-metallic material lined steel pipes, has been widely utilized for a number of years, and not just in the oil and gas industry [14]. This is because of their many benefits, including their high corrosion resistance, being lightweight, cheap cost of cycle time, and quick installation time. Glass fiber-reinforced epoxy (GRE) pipes are often made to withstand high pressure and have relatively thin walls, making them lightweight and convenient to handle and transport, which can lower installation costs [15,16]. These benefits have led to these non-metallic pipes becoming a popular choice in the structural engineering and aviation industries [17]. The extensive use of these non-metallic pipe systems necessitates the use of trustworthy testing techniques to guarantee their safety and forecast their long-term performance. However, according to some earlier studies, GRE pipes have some drawbacks related to the thermoset itself, including lower strength when compared to metallic pipes, inability to withstand high temperatures due to epoxy properties, and low corrosion resistance, which can suffer strain corrosion in acidic environments [18,19]. In order to solve this issue, this research uses high calcium pozzolanic-based geopolymer infill. The primary ingredients of cement included in today’s construction codes include natural pozzolans. Environmental policies limit their availability and, therefore, potential usage, but they are focused on limiting quarrying in order to reduce the consumption of natural resources and the degradation of the environment. Similar to other pozzolanic materials, clay-based waste has a chemical makeup that is dominated by silica, alumina, and iron oxide. In varying degrees, CaO, the alkalis Na_2_O and K_2_O, as well as a trace amount of SO_3_, are also present. Its principal crystalline phases from a mineralogical perspective include quartz, muscovite, calcite, microcline, and anorthite [20]. Fly ash, which is part of pozzolan materials, is made up of tiny fragments of ashes from the pulverized coal used in power plants. The spherical shape of fly ash aids in pipe consolidation and decreases permeability [21].

Previous researchers stated that when using pozzolan as a raw material in geopolymers, they have determined that the concentration of sodium hydroxide (NaOH) molarity should be in the range of 8 M to 16 M [22,23]. Other researchers had proven that the NaOH molarity ominously affected the mechanical properties of geopolymers and stated that the increment in compressive strength as the molarity of the NaOH increased from 8 M to 16 M [24,25]. Unfortunately, not much research was found using NaOH and geopolymer as fillers in filament winding technique. The compressive strength was increased as the concentration of NaOH increased from 6 M to 12 M. The strength then decreased, with a further increase in NaOH molarity until 16 M [26]. This may be due to the hindrance in the geopolymerization process by the excess hydroxide ion concentration, which reduces the early-stage aluminosilicate gel precipitation. As activator concentration increases, the ion species concentration also increases, thus limiting the ion’s mobility and delaying the formation of coagulated structures. This effect complements the explanation about the delay in the polymer formation as activator concentration is increased [27].

There are plenty of conditions where geopolymer materials might be exposed to extreme heat environments, such as acquaintance from fire exposure, furnaces, nuclear exposure, and from thermal processes within the geopolymer itself. In such circumstances, the appropriate support of the behavior of geopolymer materials when resolved to extreme conditions is necessary. There have been extensive investigations on the effect of thermal exposure on the mechanical properties of geopolymers, thus leading to the thermal performance of geopolymers, which is usually qualified by the modifications in the structure network of ceramic-like properties [28,29]. Quick dehydration of the weakly bound water in the matrix did not cause significant damage to the binding structure. Thus, mechanical strength was retained, and remarkable dimensional stability at high temperatures was verified [30,31]. Previous researchers stated that geopolymers maintained their amorphous structure up to 850–1300 °C [32].

According to earlier research on composite systems, which employed the formulation design of the geopolymer for the composite system, the loading of flax fibers varied from 0 to 60 wt%. [33]. Epoxy resin diglycidyl ether of bisphenol A (DGEBA) was combined with geopolymer filler because it provides low cost, simple processing, fine adhesion to numerous surfaces, and good chemical resistance for a variety of applications. One of the materials that is now receiving the most attention is thermoset-based clay composites [34]. Referring to the previous research that was carried out, very few reported on fire properties. Thus, less literature was found on reporting the interaction between the matrix and filler. With the help of this study, all of these drawbacks will be resolved, including the use of composite pipelines for aqueous liquids in the offshore oil and gas business and the production of GRE pipes with increased strength.

## 2. Materials and Experimental Details

### 2.1. Materials Selection

In this study, high calcium pozzolanic-based geopolymer filler with varied weight percentages and different sodium hydroxide (NaOH) molarity concentrations of 8 M and 12 M were mixed with glass-reinforced epoxy (GRE) pipe, which is made of diglycidyl ether of bisphenol A (DGEBA) epoxy, with a solid to liquid ratio of 1 in each case. Euro Pharma Sdn. Bhd., Penang, Malaysia provided the epoxy resin (DGEBA) (Equivalent Weight, 182–192 g/eq; Viscosity, 11,000–14,000 mPa.s; Density, 1.16 g/mL), and Dr Rahmatullah Holdings, Penang, Malaysia, provided the hardener, isophorondiamine (IPDA) (Molecular Weight, 170.3 g/mol; Viscosity, 18 mPa.s). The raw materials used to create the geopolymer filler were locally sourced from King Abdulaziz City for Science and Technology (KACST), Riyadh, Saudi Arabia, with the Blaine surface area and specific gravity ranges from 2.1 to 3.0 and 170 to 1000 m^2^/kg, respectively. An alkaline activator creates geopolymer paste to encourage silicon and aluminum atoms in the substance [35,36]. The alkaline activator liquid used in this research is a combination of sodium hydroxide and sodium silicate (Na_2_SiO_3_). Figure 1 shows the starting materials for this experiment.

### 2.2. Experimental Procedure

Four steps make up the examination of the high calcium pozzolanic-based geopolymer filler in epoxy resin for piping applications. Phase 1 is raw material characterization, as shown in Figure 2. Scanning Electron Microscopy (SEM), developed by JEOL Ltd., Tokyo, Japan, was used in this step to analyze the morphology of high calcium pozzolanic raw materials.

The morphology analysis on the microstructure of raw materials and geopolymer powder filler of high calcium pozzolanic was analyzed under 2000× and 5000× magnifications. The raw materials of high calcium pozzolanic were prepared by crushing and being sieved at 150 µm before being used for SEM analysis. While in preparation, SEM samples for the GRE pipe were cut into smaller pieces (2 cm × 2 cm) before being sent to the SEM machine. The samples were filled with high calcium pozzolanic-based geopolymer filler. Following the cutting of the GRE filled with geopolymer filler pipe, the sample was initially coated with palladium using a JEOL JFC 1600 Auto Fine Coater (Tokyo, Japan). Microstructural analysis of raw materials high calcium pozzolanic powder was spread onto double-sided carbon tape as adhesive purpose before the analysis, and the loosely held powder was removed by using a blower. Then, the sample for analysis was prepared into a 5 mm diameter cross-section and coated with palladium.

X-ray fluorescence (XRF), Bruker Malaysia Sdn. Bhd., Penang, Malaysia, with the brand name of PANanalytic PW4030, was used to analyze the chemical composition of high calcium pozzolanic. Drying the samples at 105 °C and measuring the mass loss after heating to 1000 °C were used to control the loss of ignition (LOI). The specifics of several standards measured during the analysis are included in the results, which are provided as oxides. The uncertainties were determined using the mean percentage deviation from the standards for each oxide.

Phase 2 is the synthesis of high calcium pozzolanic-based geopolymer using different molarities of sodium hydroxide, and the determination of viscosity, water absorption, compressive strength, pressure strength, and flame test of the samples. The epoxy geopolymer resin was made using a mechanical mixer and a blade stirrer, in accordance with the formulation in Table 1. With the aid of IPDA, a cycloaliphatic amine curing agent, the mixed epoxy geopolymer was cured at ambient temperature. The curing agent/hardener was added after the epoxy and geopolymer ingredients had been thoroughly homogeneous, which took about 2 h. The resins were mixed with epoxy hardener and then poured into the tank.

Phase 3 includes the development of the pipe by using epoxy filled with high calcium pozzolanic-based geopolymer filler with different filler loading. The geopolymeric resin was impregnated (“wet-out”) onto continuous glass fibers (E Type; Max Tensile Stress, 2600 MPa; Elongation of Break, 4–5%; Elastic Modulus, 7300 MPa) using a home-made “impregnation machine” as a filament winding technique. This was based on the optimal penetration of geopolymer resin into the fibers during the impregnation process, with a fixed velocity of the mandrel. To produce the necessary patterns of winding angles (90°), the winding speed (10 m/s) was adjusted. The mandrel rotational speed will affect how quickly the fiber is fed into the resin tank. It was chosen based on the best geopolymer resin impregnation into the fibers.

After applying the proper number of layers to the wounded condition, filament winding samples were left to cure at the mandrel at room temperature for 24 h. The sample was prepared for testing once the curing procedure was finished. Compressive testing and elasticity modulus measurements are just two tests done on composite structures to ascertain their mechanical properties.

The fourth phase is the durability testing of epoxy filled with different high calcium pozzolanic-based geopolymer filler loading. Three types of testing were used in this study: characterization, physical testing, and mechanical testing. The sample is characterized in terms of its shape and chemical composition. While mechanical testing involves compressive strength and hydrostatic pressure strength tests, physical testing includes viscosity, viscosity, and flame tests. The specifics of the experiment used in this study are shown in Table 2. Powder and pipe samples are the two types of specimens used in this study.

In order to explore the impact of filler loading percentage, viscosity tests were performed on both epoxy resins filled with high calcium pozzolanic-based geopolymer filler and epoxy resins without geopolymer filler with a varying weight percentage of filler from 0 wt% to 40 wt%. This is because a change in filler loading will affect the viscosity of the matrix resin by changing the contents of the solid (geopolymer filler) and liquid (epoxy resin). The average value of three samples for each parameter served as the basis for the viscosity result. The viscosity was measured using a Brookfield LVT, RVT, HBT, or HAT Dial Reading Viscometer following ASTM D4142, the industry-standard guidance for evaluating epoxy resin.

Three samples were fully submerged in distilled water at room temperature following ASTM D570 standard procedure to perform a standard test technique for water absorption. The samples were removed from the water at regular intervals, wiped with filter paper to remove surface water, and weighed on a digital scale.

The schematic diagram for testing a pipe sample for compressive strength is shown in Figure 3. Compression tests are carried out using an Instron Universal Testing Machine in accordance with ASTM D3410 standard test procedure. Each sample was positioned in the middle of the lower cross member and lower crosshead so that the load would be applied to the sample’s opposite side at a rate of 5 mm per minute. The specimens are compressed simultaneously while being positioned horizontally and vertically.

Figure 4 shows the schematic diagram for the flame test. A vertical UL 94 test was carried out using a Plastics HVUL Horizontal Vertical Flame chamber (Atlas Fire Science Products, Chicago, IL, USA) according to ASTM D635/ISO 1210 standard test procedure. Both GRE without geopolymer filler and GRE-filled geopolymer filler were evaluated under UL 94 horizontal and vertical tests involving several interacting physical processes, which are inadequately reflected in the final classification of horizontal (HB) and vertically upward (V). As the flame propagation is either HB or V, flame dilution, and hence flame dilation, is likely to have a smaller effect than energy absorption through endothermic decomposition or solid phase heat capacity, since a greater portion of the heat of combustion is fed back to the polymer.

Hydrostatic pressure leak tests were carried out using a hydrostatic-pressure machine capable of applying pressure at a uniform rate until the failure of the test specimen according to ASTM D2837 standard test procedure. The two ends of the samples were sealed using endcaps with rubber rings to prevent any longitudinal load from being transmitted to the pipe. As the pressure was increased, a special device was developed and built to measure the circumferential length. The pressure was measured using a manometer calibrated at the Pressure Vessel Testing Laboratory of the SIRIM TECH VENTURE, Penang, Malaysia. Six samples measuring 1-inch inner diameter pipe were fabricated for this test: three with geopolymer powder filler and two without geopolymer powder filler. Figure 5 shows the apparatus for the hydrostatic pressure test.

## 3. Results and Discussion

### 3.1. X-ray Fluorescence Analysis (XRF)

In X-ray fluorescence analysis (XRF), a substance that has been stimulated by being bombarded with high-energy X-rays or gamma rays emits distinctive fluorescent X-rays. Chemical and elemental analyses were performed with this test. Table 3 displays the chemical breakdown of high calcium pozzolanic materials from Saudi Arabia that were examined using X-ray Fluorescence (XRF). This high calcium pozzolanic material has a significant proportion of calcium oxide (CaO), which is considered high compared to other pozzolan materials [37,38,39,40], as stated in Table 3. According to the chemical composition results of X-ray Fluorescence (XRF), the prior raw materials utilized as filler in geopolymers have high levels of Silica (Si), Alumina (Al), Calcium (Ca), and Iron (Fe). Materials often contain silica (Si), and amorphous aluminum (Al) is a potential supply material for geopolymer production [41]. The high calcium pozzolanic was abundant in calcium oxide (CaO), silicon dioxide (SiO_2_), and aluminum oxide (Al_2_O_3_), with more than 90% of these elements being identified, as can be shown in Table 3. The presence of SiO_2_ and Al_2_O_3_ influences the performance of the geopolymer. All geopolymer materials’ properties and abilities are sufficient for use as a filler in geopolymers.

### 3.2. Surface Morphological Analysis of High Calcium Pozzolanic

The high calcium pozzolanic’s surface morphology is seen in Figure 6. It can be noticed that the high calcium pozzolanic microstructure appeared to be hollow and glassy. This unequivocally demonstrates that high calcium pozzolanic has spherical particles with smooth exterior surfaces, as claimed by earlier researchers [42]. Since the high calcium pozzolanic spherical shape was utterly broken in the raw material, there would have been some agglomeration of the fine-grained particles due to mechanical milling procedures. The smooth aluminosilicate spherical particles, also called cenospheres, are created when mineral particles are produced during coal combustion. During this process, the minerals melt to form small droplets, which quickly cool and take on a spherical shape due to surface tension forces [43,44].

### 3.3. Surface Morphology of High Calcium Pozzolanic-Based Geopolymer

Figure 7 presents a high calcium pozzolanic-based geopolymer that has been milled into fine particles (ring mil) and subjected to a 24-h curing period at an oven temperature of 80 °C. Raw high calcium pozzolanic and high calcium pozzolanic-based geopolymers’ surface morphology has changed, and this change is attributable to the reactive dissolution of SiO_2_ and Al_2_O_3_ (found in cenospheres) in alkaline solution. As a result, aluminosilicate gel will develop, acting as a precursor to the development of geopolymers [45].

Cenospheres are round hollow particles filled with gas, primarily CO_2_ and nitrogen (N_2_). According to Rahman [46], the wall of cenospheres will dissolve and release Si and Al ions when subjected to an alkaline attack. Due to the high calcium (Ca) content of raw high calcium pozzolanic materials, during the geopolymer’s dissolving process, the CaO content of these materials was affected by the reaction products from the alkali solution and changed into active species of Ca^2+^ and O^2−^. A geopolymer composite was created when Ca^2+^, Si^4+^, and Al^3+^ ions reacted with the OH ions in the alkali solution. At this stage, the Ca content in high calcium pozzolanic began promoting the optimum reaction within the geopolymer system. High calcium pozzolanic fine particle size contributes to a faster initial setting time for geopolymer and increases compressive strength. Geopolymer paste with high flowability is produced due to the spherical shape’s ability to facilitate easier particle sliding between one another. As a result, less liquid would be needed to create geopolymer paste. This is important because, as earlier research has demonstrated, the porosity decreases as less water is used [47,48].

### 3.4. Viscosity Analysis Test

A standard test method for Brookfield Testing for viscosity was done following ASTM D445 standard, the industry-standard procedure for evaluating epoxy resin [49]. According to the standard, the time it takes for a volume of liquid to flow under gravity through a calibrated glass capillary viscometer is used to estimate the kinematic viscosity. The dynamic viscosity can be calculated by dividing the measured kinematic viscosity by the liquid’s density [49].

From Figure 8, it is clearly shown that the viscosity of the epoxy resin increases with the increase of wt% geopolymer filler in the epoxy resin. The epoxy resin’s viscosity shows the lowest viscosity at 5766.67 MPa.s, while epoxy with the 40 wt% of geopolymer filler gives the highest viscosity at 17,302.00 MPa.s.

In addition to geopolymer filler wt%, the viscosity of epoxy geopolymer continued to increase by increasing the geopolymer filler. It can be inferred that the geopolymer filler was responsible for the viscosity of the epoxy resin. However, the geopolymer filler did not increase the total weight, but the role of the geopolymer filler was to decrease the use of epoxy resin, thereby increasing the number of micro clay-based fillers inside the epoxy resin, which led to the enhancement of mechanical properties. Comparing the high and low viscosity epoxy composites showed that the dispersion of low viscosity epoxy composites was better than the high viscosity epoxy composites due to the different polymer viscosity [50]. At the filler loading from 0 wt% to 30 wt% of geopolymer filler loading, the viscosity of the matrix was still low. The high viscosity of the matrix due to the addition of more filler caused the presence of porosity in the composites and, in turn, caused the wettability between fiber glass and matrix to become poor. The presence of more porosity reduced the compressive strength of the GRE pipe filled with geopolymer filler composites. It is clearly shown that the increasing geopolymer filler up to 40 wt% of filler loading decreased the compressive strength test of the samples in this study. This substantial drop in strength may be related to the increase in the viscosity of the matrix, which reduced filler wetting and led to the emergence of many flaws and gaps within the composites [51].

### 3.5. Water Absorption Test

Three samples were submerged in distilled water at room temperature to perform a standard test technique for water absorption following ASTM D570 [52]. The samples were removed from the water at regular intervals, dried off with filter paper to remove surface water, and weighed on a digital scale. Water absorption was utilized to calculate the amount of water absorbed under particular circumstances.

From Figure 9, it can be seen that the percentage of water absorption decreases with the increase of geopolymer content (filler loading) for both samples 8 M and 12 M. The percentage of water absorption for samples high calcium pozzolanic-based geopolymer for 8 M result shows 10 wt% of geopolymer content gives the higher water absorption compared to samples with 12 M of NaOH, while the percentage of water absorption for high calcium pozzolanic-based geopolymer with 40 wt% of geopolymer filler 8 M shows the lowest water absorption. This is due to the chemical composition in the high calcium pozzolanic, which contains higher CaO and lower OH. This can drive the hydrophobic reaction from the material itself.

In general, water uptakes were influenced by the character of the matrix resin and the glass fibers of the hydrophilic character, the adhesion between the matrix resin itself and the glass fiber, and the presence of void or agglomeration in the composite [53]. The alleviation in the crosslinking density seems to be one factor controlling the water’s absorption. Thus, the reduction of the water absorption when the geopolymer filler content increases is due to the high calcium (Ca) content in the pozzolanic material, which contains more clay-based materials properties than cementitious material.

### 3.6. Surface Morphological of Glass-Reinforced Epoxy Filled with Geopolymer Filler

In order to study the adhesion between the glass fiber and epoxy geopolymer filler matrix, the images were investigated. Figure 10 shows that the matrix resin without any geopolymer filler covering the glass fiber seems almost similar. According to the microstructure analysis of the GRE pipe, interfacial deboning between the glass fibers and the surrounding matrix is the main feature rather than the matrix failure [54].

Figure 11 shows the SEM micrographs of GRE filled with high calcium pozzolanic-based geopolymer pipe with different wt% of filler loading. At the optimum weight percentage of filler loading, 30 wt% showed better resin and filler distribution, contributing to the highest strength of samples produced. At the highest percentage of filler loading, 40 wt%, it can be seen that the matrix resin contains too much filler and less interaction with the fiber itself, and thus will weaken the bonding between matrix resin and fiber glass. A GRE pipe interphase is the region between glass fiber and matrix with different chemical and physical properties from those of glass fiber and matrix. It is generally recognized that even in the presence of the filler in matrix resin, the fiber-to-matrix interface is still the weakest portion of the fiber-reinforced resin [55].

However, as can be seen in Figure 11c,d when the resin was damaged during the filament winding process, this caused many voids, and agglomerations of the geopolymer filler itself were created during the mixing process. Consequently, the internal structure of the composites deteriorated. As a result, the viscosity of the composite increased, and the strength dramatically dropped, as shown in Figure 8. Figure 10 showed less matrix resin adhered to the glass fiber than the composite pipe when high calcium pozzolanic-based geopolymer filler was applied (Figure 11a–d). This might result from a stronger attraction between the surface of the fibers and the high calcium pozzolanic-based geopolymer filler.

Additionally, the SEM revealed a more brittle surface in the glass-reinforced epoxy composite without any geopolymer filler content and a significantly rougher surface for the geopolymer filler. This is consistent with their increased strength, as determined by the compression testing (Figure 12). The compressive properties investigation revealed the GRE pipe’s mechanical characteristics, including geopolymer filler.

### 3.7. Compression Analysis Test

Compression tests were performed using the Instron Universal Testing Machine following ASTM D3410 [56]. To perform a compression test, a cylindrical hollow specimen is deformed into a thinner but larger diameter cylinder hollow. It is an easy way to ascertain the stress–strain response. The 0–40 wt% high calcium pozzolanic-based geopolymer filler loading for compressive strength, compressive strain, and compressive modulus of elasticity are shown in Table 4.

Figure 12 depicts the usual compression strength test graph for the glass-reinforced epoxy (GRE) pipe without filler and the glass-reinforced epoxy (GRE) pipe loaded with a high calcium pozzolanic-based geopolymer filler that is 10 wt% to 40 wt%.

Figure 12 clearly illustrates that the compressive strength of GRE pipes with high calcium pozzolanic-based geopolymer filler for both 8 M and 12 M of NaOH improves as the geopolymer filler percentage increases up to 30 weight percent. However, at 40 weight percent of geopolymer filler, the compressive strength of GRE pipe filled with high calcium pozzolanic-based geopolymer filler for both 8 M and 12 M somewhat decreases. This can be because the increased viscosity of the geopolymer filler, when combined with the epoxy, results in challenging workability between the epoxy and filler. The compressive strength for 12 M is still higher than the GRE pipe without any geopolymer filler, even if the compressive strength of the GRE pipe with high calcium pozzolanic-based geopolymer filler decreases when added with increased filler loading.

The rule of mixture theorem was used to support the claim that adding microparticles makes an epoxy matrix more rigid [57]. The matrix and the interface are two variables that can be justified as influencing how they behave when they are strong. The high calcium pozzolanic-based geopolymer affects the matrix properties, and it was claimed that adding cement to a polymer increased its impact strength because the geopolymer filler created a tortuous fracture path, and the cementitious platelets prevented the development of microcracks [58,59,60,61]. The fact that the compressive strength of a polymer composite decreases with the strength of the contact between the matrix and the fiber at this point is indisputable. This could occur due to the usage of lower molarity NaOH and the smaller, more agglomerated, and disordered form of clay.

It is clear from the enhanced compressive strength that the geopolymer filler and epoxy hardener interact strongly inside the polymer chains. However, the ability and capacity of relocating load and plastic deformation between particles and matrix interface may be controlled by the reduction of compressive strength in increased containment of high calcium pozzolanic at 40 wt%.

### 3.8. Hydrostatic Pressure Strength/Burst Analysis

The hydrostatic pressure strength test was performed according to the ASTM D2837 Hydrostatic Pressure Test. The hydrostatic pressure or burst test results of control samples (GRE pipe without geopolymer filler) compared with the best compressive strength for GRE filled with high calcium pozzolanic-based geopolymer pipe (12 M NaOH concentration at 30 wt% of filler loading) are shown in Figure 13. The samples for each pipe (GRE without any geopolymer filler and GRE-filled high calcium pozzolanic-based geopolymer filler) were produced with two patterns of filament winding types—hoop pattern (90° angle) and helical pattern (55° angle). As can be seen in Figure 13, it is clearly shown that the samples with hoop pattern give higher pressure strength (225 bar and 261 bar) compared to the helical pattern (181 bar and 233 bar) for each control sample and GRE filled with high calcium pozzolanic-based geopolymer pipe.

Generally, the pressure strength test increased when geopolymer filler was added to this composite system. This is attributed to the increased mechanical strength of the pipes following the plasticization of the matrix, which strengthens the interfacial fiber–matrix interface and the stress transferred from the epoxy matrix resin to the high calcium pozzolanic-based geopolymer filler [62]. The GRE filled with high calcium pozzolanic-based geopolymer with hoop pattern of filament winding pattern shows the highest-pressure strength (261 bars) compared to the control sample with the same pattern type, which shows only 225 bars. This indicates that geopolymer filler can improve the strength properties of the pressure of the existing GRE pipe. The testing results proved that the geopolymer filler significantly affects the pressure resistance of the GRE pipes. This is happening because energy is absorbed in the development of plastic deformation of the matrix material, debonding at the matrix/reinforcement interface, and in the fracture of reinforcing material. Thus, the geopolymer filler is applicable and suitable to use in matrix resin with fiberglass pressure pipes, according to the results of the hydrostatic pressure leak tests.

### 3.9. Flame Retardancy and Burning Behaviors

To explore the flame retardancy of the GRE and GRE filled with high calcium pozzolanic-based geopolymer filler samples, horizontal and vertical burning ratings (UL-94) of the samples without geopolymer filler and GRE filled with high calcium pozzolanic-based geopolymer filler with different filler loading samples were measured, and the results are presented in Table 5. Figure 14 shows a bigger change in shape for GRE without geopolymer filler after 10 s of exposure to fire for both horizontal and vertical sides compared to GRE filled with high calcium pozzolanic-based geopolymer filler. It can also be clearly seen that the epoxy for the GRE without geopolymer filler was melted down after being exposed to direct fire. Contrarily, for high calcium pozzolanic-based geopolymer filler pipe samples, the epoxy does not melt and remains in its shape. However, some small cracks have occurred on the surface of the pipe samples.

The UL-94 ratings of the GRE without filler, GRE with 10–40 wt% of high calcium pozzolanic-based geopolymer filler loading samples were, respectively: V2, V1, V1, V1, and V1. Only GRE without geopolymer filler samples exhibited dripping, while GRE with geopolymer filler exhibited no dripping. With an increase in the geopolymer filler loading content, the self-extinguishing time of the sample was significantly reduced.

This is due to the thermal resistance and thermal stability of the geopolymer filler that hindered the heat transfer from the fire to the epoxy. Small cracks emerged after the collapse of a covering flame zone in the GRE filled with high calcium pozzolanic-based geopolymer filler at the conclusion of the experiment. The easiest way to explain the charring behavior is to contrast it with what is seen in wood burning. At the conclusion of the experiment, the stiff, black residue left over from the combustion of GRE without any geopolymer infill begins to glow. At first look, the visual observations for GRE with and without geopolymer filler pipe samples appear to be fairly similar, however the flame zone appears to have been substantially reduced during the experiment, and the glowing of the char at the conclusion of the experiment has been altered slightly.

According to Figure 14B, the minor surface crack on the pipe sample of GRE is filled with high calcium pozzolanic-based geopolymer filler is made up of hundreds of wormlike structures that rise to the surface. This creates a much larger surface area than the GRE without geopolymer filler, which has melted down. The addition of high calcium pozzolanic-based geopolymer filler results in a significant reduction of the melted epoxy. The flame extension is reduced drastically for GRE filled with high calcium pozzolanic-based geopolymer filler in comparison to the GRE without geopolymer filler. For GRE without geopolymer filler, the transition between burning with a flame and glowing without a flame occurs almost continuously during the experiment. Some of this leftover epoxy slips down softly, especially for GRE without geopolymer filling. Therefore, the sample mass during combustion is affected by a second mass loss in addition to the generation of volatile breakdown products.

## 4. Summary and Conclusions

The development of glass-reinforced epoxy (GRE) pipes filled with high calcium pozzolanic-based geopolymer on epoxy resin with varying weight percentages and sodium hydroxide (NaOH) molarities of the geopolymer material may be inferred as the conclusion.

The experimental result shows the performance of the product through the surface analysis, surface morphology, mechanical testing, and thermal/fire resistance testing. The samples from GRE pipe with high calcium pozzolanic-based geopolymer of 30 wt percent, 12 M, show the maximum strength relative to the other samples, whereas GRE pipe without any geopolymer filler shows the lowest compressive strength, according to the results of the compression tests and the pressure strength test.GRE-filled high calcium pozzolanic-based geopolymer pipe samples show high thermal stability during thermal/fire resistance tests when compared to GRE without geopolymer filler pipe samples.The geopolymer made from waste materials has greater potential to be used as a matrix filler in composite materials with glass fiber when using a filament winding technique because of its good qualities.By using a filament winding process, geopolymer composite can be used as a filler in piping systems, which is not only more ecologically friendly but also lower in cost to produce.

## Figures and Tables

**Figure 1 materials-15-06495-f001:**
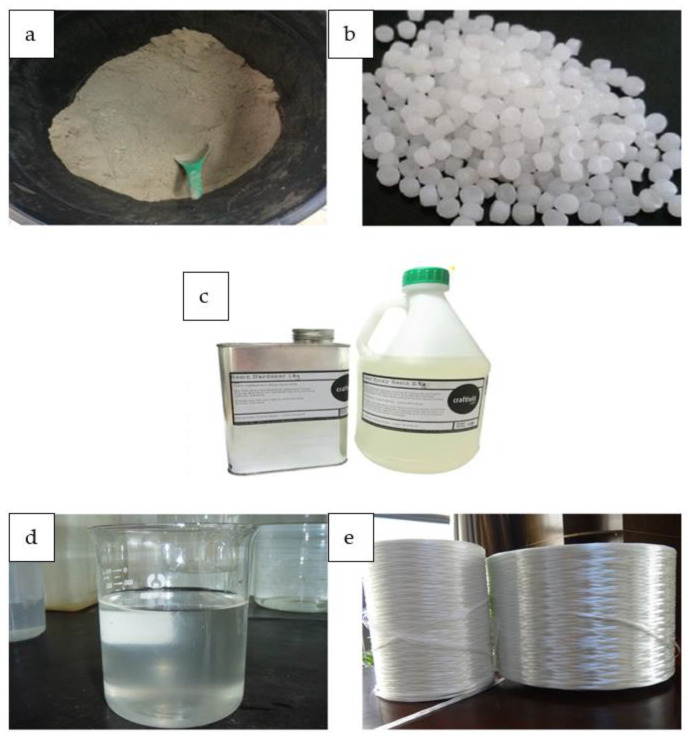
Starting materials. (**a**) High calcium pozzolanic, (**b**) sodium hydroxide pellet, (**c**) epoxy resin (DGEBA) and epoxy hardener (IPDA), (**d**) sodium silicate liquid, and (**e**) fiber glass type E.

**Figure 2 materials-15-06495-f002:**
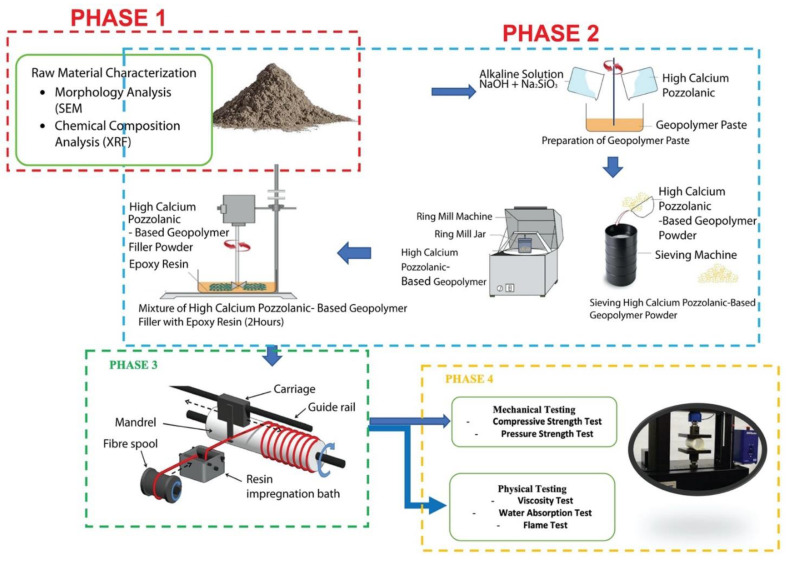
Process flow of the product development and characterization for epoxy filled with geopolymer composite pipe via filament winding.

**Figure 3 materials-15-06495-f003:**
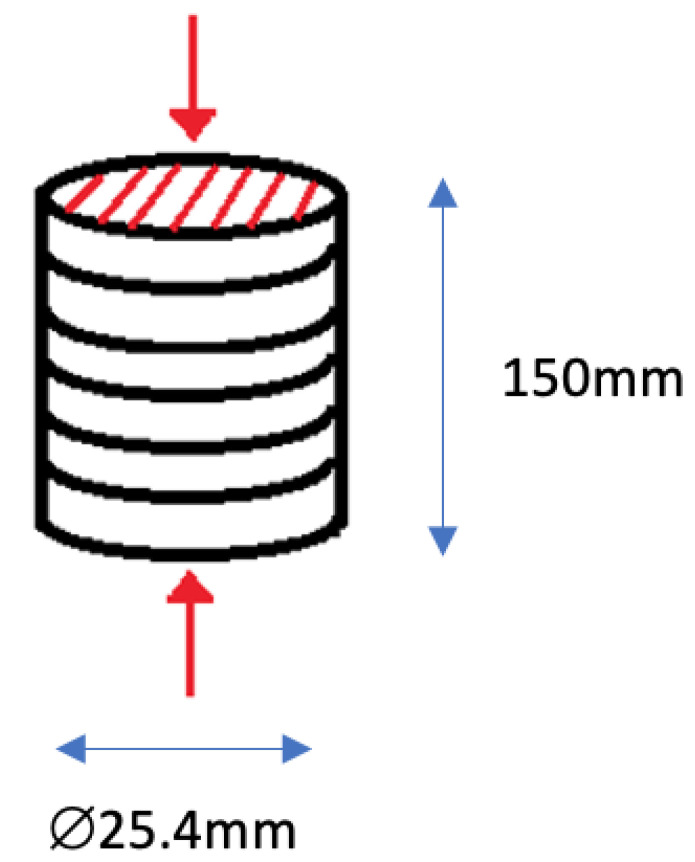
Schematic diagram of testing pipe sample on vertical position.

**Figure 4 materials-15-06495-f004:**
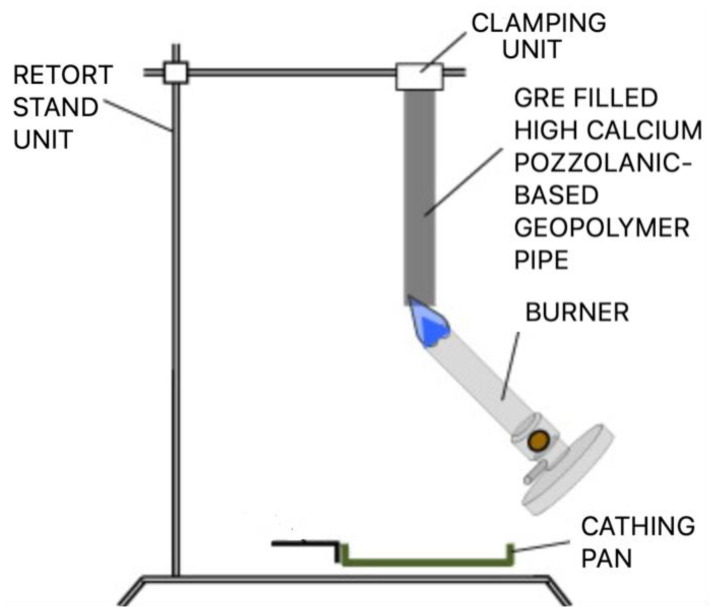
Schematic diagram of flame test.

**Figure 5 materials-15-06495-f005:**
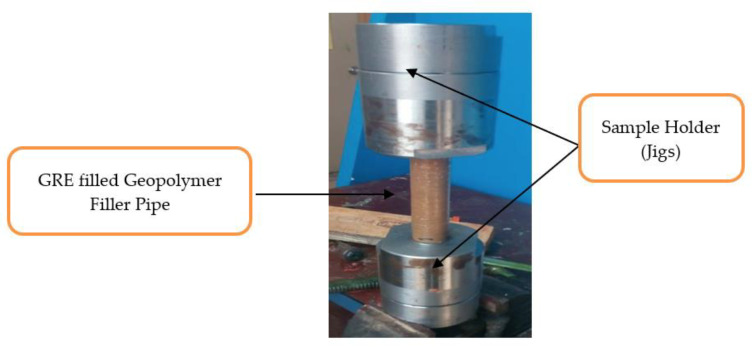
Hydrostatic pressure leak test.

**Figure 6 materials-15-06495-f006:**
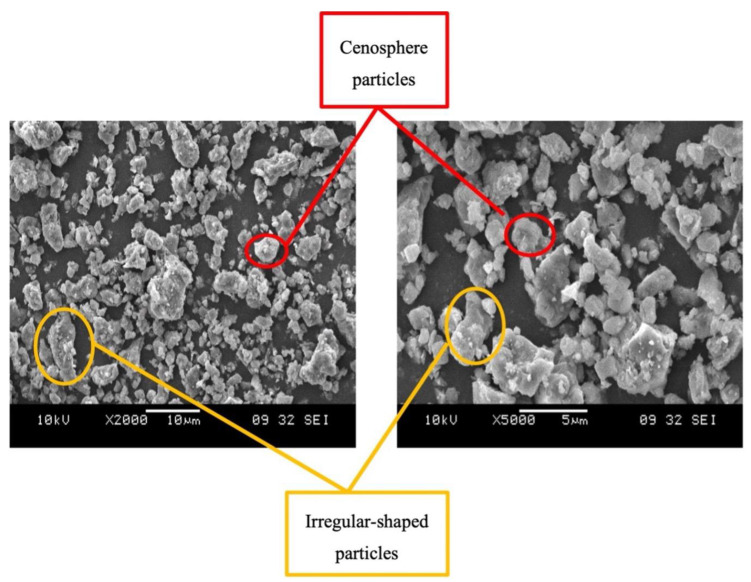
SEM micrographs of the raw high calcium pozzolanic at 2000× and 5000× magnification.

**Figure 7 materials-15-06495-f007:**
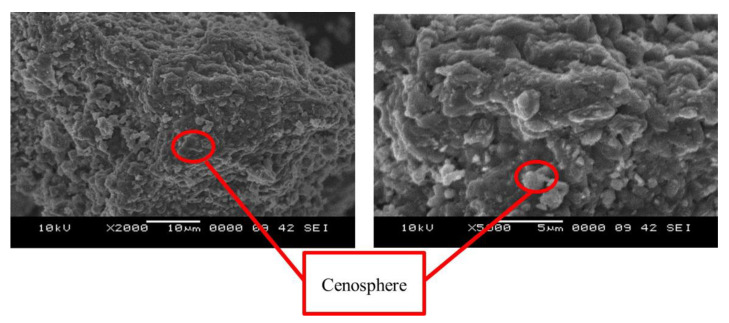
SEM micrographs of the high calcium pozzolanic-based geopolymer at 2000× and 5000× magnification.

**Figure 8 materials-15-06495-f008:**
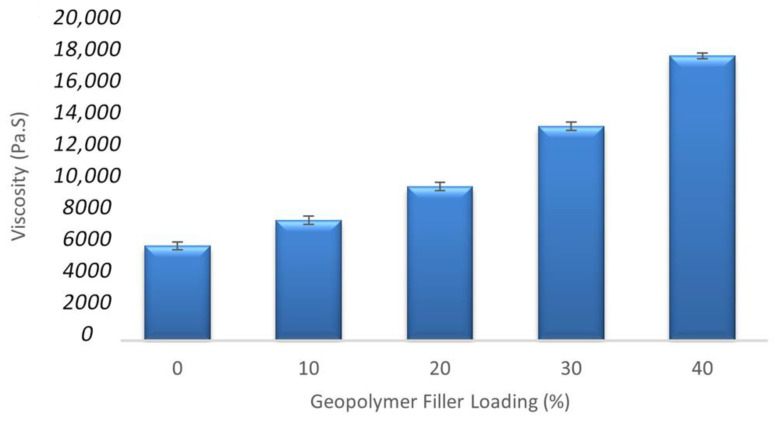
Comparison graph of liquid viscosity between samples.

**Figure 9 materials-15-06495-f009:**
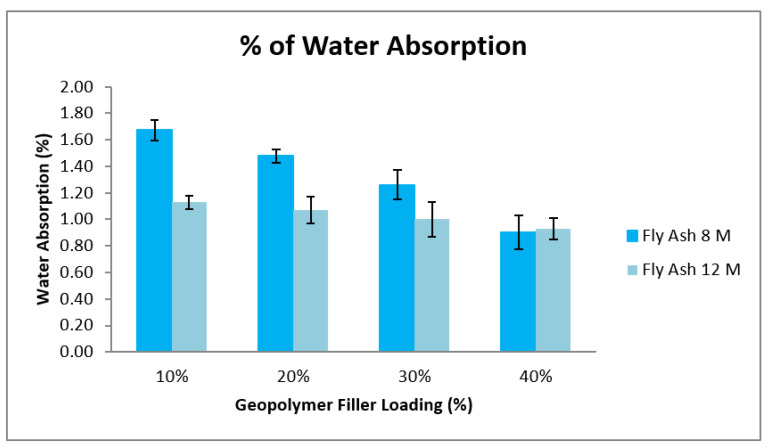
Comparison graph of % water absorption between samples.

**Figure 10 materials-15-06495-f010:**
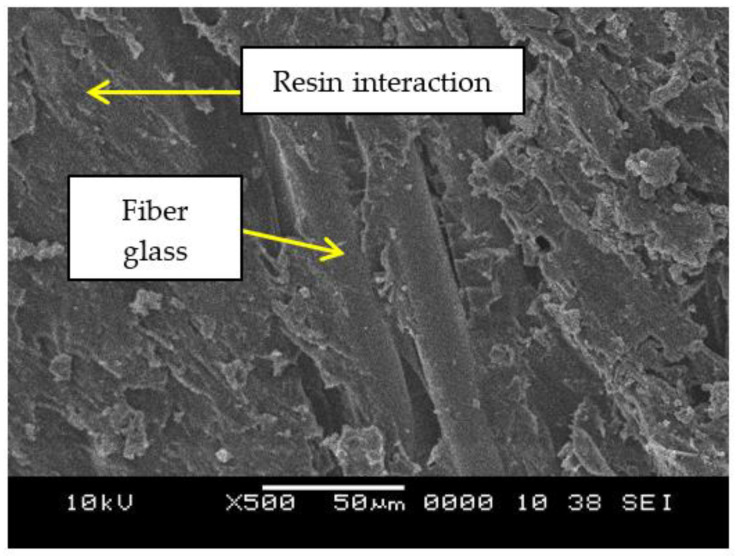
SEM images of control sample GRE pipe without any geopolymer filler.

**Figure 11 materials-15-06495-f011:**
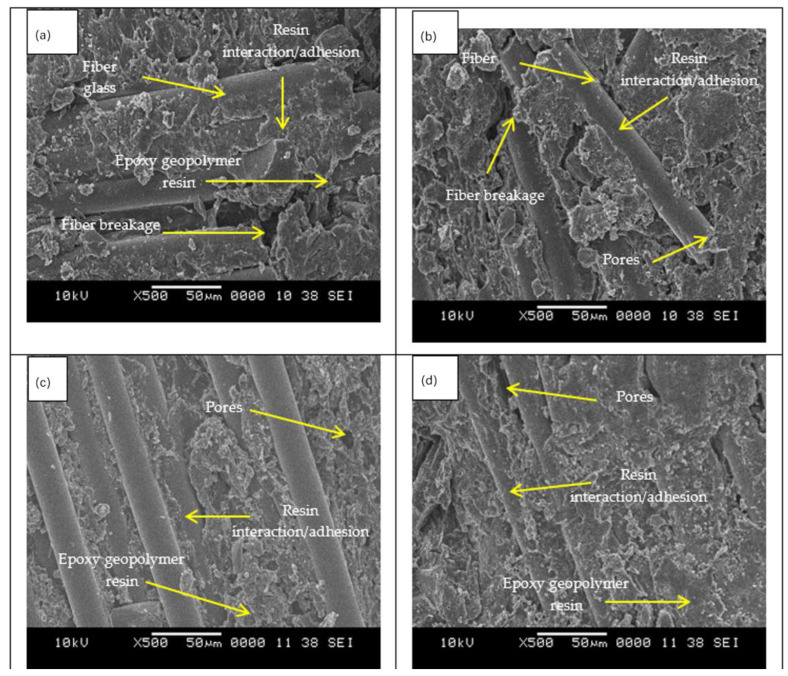
SEM images of GRE-filled high calcium pozzolanic-based geopolymer filler pipe (**a**–**d**) with different wt% of filler loading. (**a**) 10 wt%, (**b**) 20 wt%, (**c**) 30 wt%, (**d**) 40 wt%.

**Figure 12 materials-15-06495-f012:**
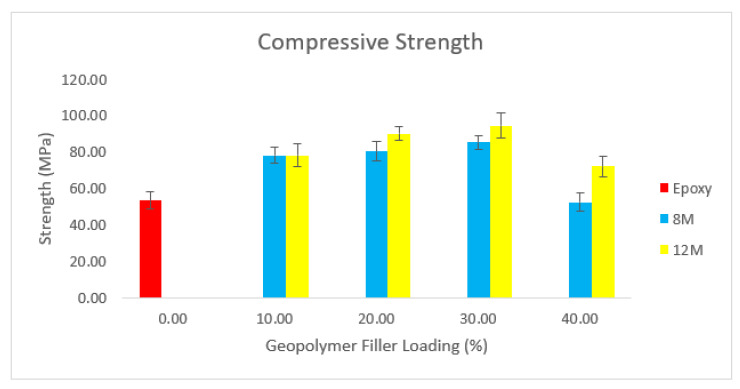
Compressive strength of GRE pipe with 0–40 wt% geopolymer filler loading.

**Figure 13 materials-15-06495-f013:**
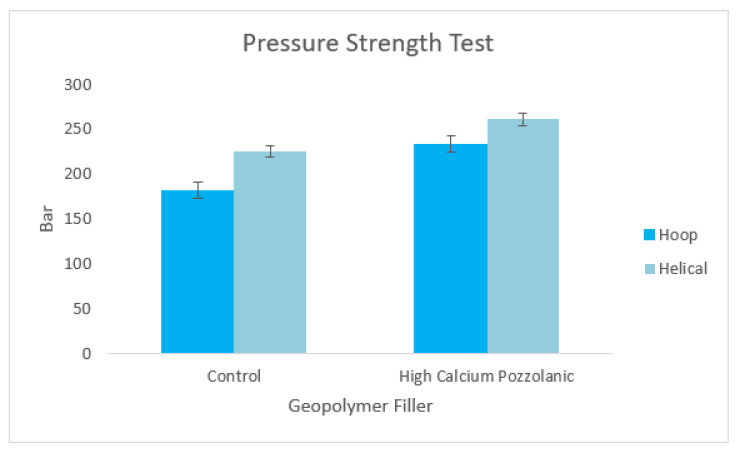
Burst test of the control sample and GRE filled with high calcium pozzolanic-based geopolymer pipe.

**Figure 14 materials-15-06495-f014:**
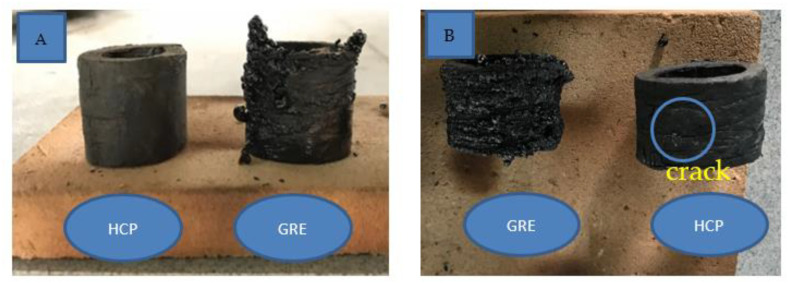
Heat resistance properties of (**A**) horizontal and (**B**) vertical of GRE pipe with high calcium pozzolanic-based geopolymer filler and GRE without geopolymer filler.

**Table 1 materials-15-06495-t001:** Epoxy geopolymer resin formulation design.

Raw Material	Epoxy + Hardener (%)	Geopolymer (%)
High Calcium Pozzolanic	100	0
	90	10
	80	20
	70	30
	60	40

**Table 2 materials-15-06495-t002:** Experiment details.

Specimen	Test	Standard	Unit	Size of Samples	Age of Test	No. of Samples
Epoxy filled Geopolymer Filler	Viscosity Test	ASTM D445	Pa.s	N/A	2 h	3
GRE filled Geopolymer filler	Water Absorption Test	ASTM D570	%	50.8 mm (Ø), 3.2 mm thickness	28 Days	3
GRE filled Geopolymer filler	Compressive Strength Test	ASTM D3410	MPa	25.4 mm (Ø)	28 Days	3
GRE filled Geopolymer filler	Pressure Strength Test	ASTM D2837	Bar	25.4 mm (Ø)	28 Days	3
GRE filled Geopolymer filler	Flame Test	ASTM D3801	N/A	25.4 mm (Ø)	28 Days	3

**Table 3 materials-15-06495-t003:** Chemical composition of high calcium pozzolanic.

Oxide Composition	High Calcium Pozzolanic (%)
SiO_2_	9.07
Al_2_O_3_	3.1
CaO	80.59
Fe_2_O_3_	5.19
MgO	0.84
ZrO_2_	0.032
TiO_2_	0.39
K_2_O	0.465
V_2_O_5_	0.030

**Table 4 materials-15-06495-t004:** Compression properties of GRE pipe without geopolymer filler and GRE filled with geopolymer filler.

Compressive Strength (MPa)					Compressive Strain (%)				Modulus Elasticity (MPa)			
	10	20	30	40	10	20	30	40	10	20	30	40
Epoxy	53.36				0.04				1681.28			
8 M	78.30	80.50	85.48	52.55	30.03	30.03	22.69	0.10	2166.61	2199.71	1984.72	1874.41
12 M	78.33	90.19	94.64	72.13	0.04	0.06	0.06	0.05	2564.09	2699.54	2373.58	2183.51

**Table 5 materials-15-06495-t005:** UL-94 results of GRE without filler and GRE with high calcium pozzolanic-based geopolymer filler samples.

Samples	UL-94 Rating	Dripping
GRE without Filler (Vertical)	V2	Yes
GRE without Filler (Horizontal)	HB	Yes
GRE with 10% Geopolymer Filler (Vertical)	V1	No
GRE with 20% Geopolymer Filler (Vertical)	V1	No
GRE with 30% Geopolymer Filler (Vertical)	V1	No
GRE with 40% Geopolymer Filler (Vertical)	V1	No
GRE with Geopolymer Filler (Horizontal)	HB	No

## Data Availability

Not applicable.

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
