# Peer review of "Interaction of Geopolymer Filler and Alkali Molarity Concentration towards the Fire Properties of Glass-Reinforced Epoxy Composites Fabricated Using Filament Winding Technique"

_materials, 2022, doi:10.3390/ma15186495_

Round 1

Reviewer 1 Report

·        - The structure of the paper could be improved to facilitate the comprehension of the performed analyses and experiments. In particular, the authors could add, in the introduction, a paragraph where the paper contents can be briefly presented.

·        - The resolution of Figure 1 should be improved and the font size should be bigger.

·        - Figures 2, 3 and 4 can be grouped into one Figure.

·        - Please check for typos and for misuse of technical English.

Author Response

Dear Prof,

Reviewer 2 Report

-The paper is full of scientific and grammatical mistakes

-The chemical composition of fly ash is completely wrong

-The SEM images of raw fly ash are completely wrong

-The results of the experimental are suspicious

-The title is too long

-It is not recommended to include any abbreviation in the paper title “Interaction of……GRE….”

-The abstract is too long and should be condensed

-The paper is fill with self-citations and some of them have nothing to do with the subject of the paper. This not a good scientific sign. The authors must keep only the necessary self-citations. In addition, it is better to variety the references

-The paper is full of mistakes especially those related

-English should be checked with an English native or a special editor

-Page 1, “between fly ash-based geopolymer matrixes” should be “between fly ash-based geopolymer matrices”

-Page 1, please replace “created” with another suitable verb “winding filaments, composite samples were created“

-Page 1 “are” should be “were” in this sentence “values of compressive strength are 94.64 MPa and compressive modulus are 2373.58 MPa“ and in other similar sentences

 -Page 1, avoid using future tense “analysis of fly ash-based geopolymer filler will make it easy to obtain the maximum”

-Page 2, “strong application” OR “strong applications”?

-Page 2, this sentence “is no car-bon dioxide (CO2) emission” is wrong, of which there is CO2 emissions, but lower than Portland cement  

-Page 2, “in a source material of geological origin or in by-product materials such as fly ash, kaolin,” OR “in a source material of geological origin or in by-product materials such as fly ash, metakaolin,”?

-Page 2, reference [2] is inappropriate in this sentence “in a source material of geological origin or in by-product materials such as fly ash, kaolin, and rice husk ash to produce binders [2]”

-Page 2, why the authors wrote capital letter “Due to”

-Page 2, “creation” is not suitable in this sentence “made toward the creation of inorganic geopolymers”

-Page 2, it is too much to cite 6 references for this sentence “The literature has several publications on the synthesis, characteristics, and uses of geopolymers [3–8]”, the authors can cite one or two reference(s) related to review paper, of which any review paper contains lots of research paper

-Page 2, “Since clay materials are naturally”, but there are also waste materials

-Page 2, check English roles “The compressive strength increases as NaOH concentration is increased from 6M to 12M”

-Page 2, please explain why “The strength then decreased with a further increase in NaOH molarity until 16M [28].”

-Page 3, “There are plenty of application and condition” OR “There are plenty of applications and conditions”

-Page 3, long sentence “There are plenty of application and condition where geopolymer materials might be exhibit to extreme heats environments such as acquaintance from the fire exposure, fur-naces, nuclear exposure, and also from thermal processes within the geopolymer itself”

-Page 3, it is too much to cite 5 references for this sentence “in the structure network of ceramic-like properties [29-33]”, the authors can cite one or two reference(s) related to review paper(s)

-Page 3, avoid using future tense “loading, geopolymer will show declined”

-Page 3, this sentence “Certainly, under thermal loading, geopolymer will show declined its strength” is not accurate, of which the compressive strength of fly ash geopolymer may increase after exposure to 800 and 1000 C, but this depend on the initial curing conditions

-Page 3, modify “design of the geopolymer is in reference to the composite “

-In Section 2, please include photos for all starting materials

-In Section 2.1, the properties of the raw materials should be included. For example, the Blaine surface area and specific gravity of fly ash should be include. The full properties of GRE should be included

-Page 3, are you sure that “Raw materials fly ash was prepared by crushing and sieved at 150 μm before used”, already the flay ash is available in its powder form

-Page 3, it is too much to use pieces of 2 cm for SEM “SEM samples for GRE pipe were divided into smaller pieces (2 cm x 2 cm) “

-Page 4, the curing conditions should be included “the mixed epoxy geopolymer was cured”

-Page 5, the explanation of phase 3 is not clear

-In Table 2, are you sure that the used size of the samples satisfy the used standard?

-In Table 2, the full dimensions of the samples should be included in Table 2 instead of included “I inch pipe (Φ)”

-In Table 2, the dimensions should be in millimeter

-In Fig. 2, please include the dimensions of the cylinder

-The mix design should be included in a separate Table

-The chemical composition of fly ash presented in Table 3 is completely wrong

-The SEM images of raw fly ash are completely wrong

-The SEM images of fly ash geopolymer are strange

**Sorry, the authors cannot complete his detailed review because the paper is full of mistakes**

Author Response

Dear Prof, 

Thank you very much for your valuable comments to help us to improve this manuscript. 

Reviewer 3 Report

For the investigation ofthe the Properties of GRE Composites, the proportion of Geopolymer Filler and Alkali Molarity Concen-tration are two critical parameters and should be paid more attention. In this work, the authors investigate the effect of "Interaction of Geopolymer Filler and Alkali Molarity Concen-tration Towards the Fire Properties of GRE Composites Fabri-cated Using Filament Winding Technique" , it is of great significance to understand the high-temperature behavior of geopolymer matrixes. This paper can be published in this journal; however, some revisions should be done before that.

-It should have been interesting to elaborate the significance, meaning or applications of the study in introduction and conclusion parts, respectively.

-What was the reason for using Geopolymer Filler in the analyzed compositions?

-Better to clarify the meaning of every abbreviation appeared firstly.

-In 2.2 standards are difficult to find. Can the procedures shortly be described?

-In 3.1, is any particular properties of FLY ASH to emphasize? if not, just put Part3.1 in 2.1.

-The discussion is a little disappointing. It deserves to make more links between the exposed data as well as with the results of the literature. 

-The conclusion derived is redundant, just list it one by one.

Author Response

(The authors gave the same response as above.)

Round 2

Reviewer 2 Report

The paper is still fill with mistakes

The authors performed some comments and ignored the others

Author Response

Dear Reviewer, 

Please refer to the attached file for the response. 

Thank you

Reviewer 3 Report

There are no other issues with the manuscript expect language, it must be improved and try to avoid ambiguity. 

Author Response

Dear Reviewer, 

Round 3

Reviewer 2 Report

  • The paper is still fill with scientific mistakes, for example, and not as a limitation:

®    The chemical composition of fly ash is completely wrong

®    The SEM images of raw fly ash are completely wrong

  • These scientific mistakes lead to a loss of confidence and credibility in the results included in the paper
  • The authors performed some comments and ignored the others
  • The author took more than one opportunity to avoid errors and improve the research but could not, so I must reject the paper

Author Response

Dear Reviewer,

Kindly refer to the attached file given. 
